# RAGDP: Retrieve-Augmented Generative Diffusion Policy

## Abstract

Diffusion Policy has attracted attention for its ability to achieve significant accuracy gains in a variety of imitation learning tasks. However, since Diffusion Policy relies on the Diffusion Model, it requires multiple denoising steps to generate a single action leading to long generation times. To address this issue, methods like DDIM and Consistency Models have been introduced to speed up the process. While these methods reduce computation time, this often comes at the cost of accuracy. In this paper, we propose RAGDP, a technique designed to improve the efficiency of learned Diffusion Policies without sacrificing accuracy. RAGDP builds upon the Retrieval-Augmented Generation (RAG) technique, which is commonly used in large language models to store and retrieve data from a vector database based on encoded embeddings. In RAGDP, pairs of expert observation and actions data are stored in a vector database. The system then searches the database using encoded observation data to retrieve expert action data with high similarity. This retrieved expert data is subsequently used by the RAGDP algorithm to generate actions tailored to the current environment. We introduce two action generation algorithms, RAGDP-VP and RAGDP-VE, which correspond to different types of Diffusion Models. Our results demonstrate that RAGDP can significantly improve the speed of Diffusion Policy without compromising accuracy. Furthermore, RAGDP can be integrated with existing speed-up methods to enhance their performance.

## 1 Introduction

In the effort to teach behaviors to intelligent agents, imitation learning has been utilized to solve various tasks (Schaal, 1999; Osa et al., 2018). With the success of Diffusion Models in other fields, researchers has been experimenting using these models for imitation learning showing excellent results (Team et al., 2024; Chi et al., 2023; Ze et al., 2024; Reuss et al., 2023; Chen et al., 2024), with Diffusion Policy (Chi et al., 2023) achieving state-of-the-art performance in Behavior Cloning.

Despite these advancements, Diffusion Policy's reliance on Diffusion Models introduces a significant computational cost. The core challenge stems from the need to perform sequential denoising of full Gaussian noise to generate a single sample, which greatly increases inference time. For instance, Diffusion Policy operates using Denoising Diffusion Probabilistic Models (DDPM) (Ho et al., 2020), which require approximately 100 iterations of denoising to generate an action from Gaussian noise. While reducing the number of denoising steps can improve speed, it often leads to a trade-off in accuracy, as noise cannot be fully eliminated in fewer steps.

To further enhance speed, methods that reduce the number of required steps have been explored (Song et al., 2022; Salimans & Ho, 2022; Song et al., 2023; Kim et al., 2024). These approaches show an increase in generation speed, but involve inherent trade-offs in quality, for example in multi-stage tasks. A reduction in accuracy, even if small, can have compounding effect in imitation learning due to the covariant shift leading to sub-optimal policies (Ross et al., 2011; Rajaraman et al., 2020). This limitation may further restrict applicability in domains that require high-precision movements, such as robotics (Ke et al., 2021).

Recent advances in retrieval-augmented diffusion models, such as Retrieval-Enhanced Asymmetric Diffusion (READ) (Oba et al., 2024) for motion planning, Retrieve-Augmented Generation (RAG) (Lewis et al., 2021) for text generation, and ReDi (Zhang et al., 2023) for efficient image

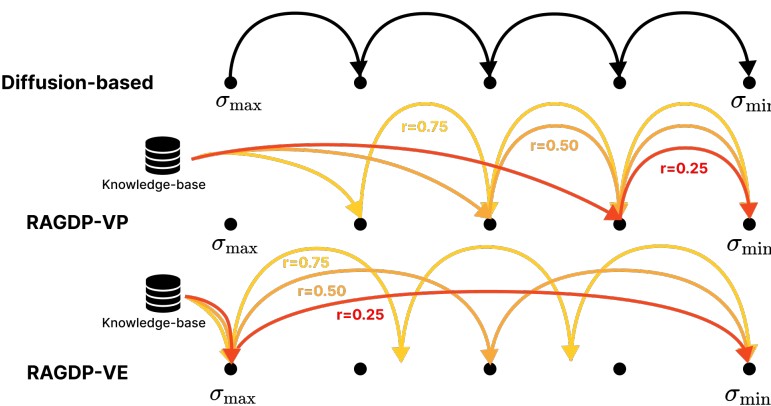

Figure 1: **Diffusion-based Policies and RAGDP** RAGDP can generate actions with two methods, RAGDP-VP and RAGDP-VE; it can obtain neighborhood values from the knowledge-base and adjust the generation speed by parameter $r$.

generation, demonstrate retrieval in enhancing diffusion processes. However, these models focus on refining trajectories or text generation, lacking the capability to generate action policies for imitation learning. Our method addresses this by introducing retrieval-based expert trajectories into a diffusion framework for action policy generation. We focus on speeding up policy inference while maintaining high-quality action generation, which is important for applications dependent on the inference time.

We introduce Retrieve-Augmented Generation Diffusion Policy (RAGDP) accompanied with two action generating methods, RAGDP-VP and RAGDP-VE used to accelerate the denoising process in diffusion models for imitation learning. Analogical to RAG (Lewis et al., 2021), we store expert observations and action data in a vector database. When denoising the current action, the model can search the available database based on its encoded observation data to retrieve actions with high similarity to generate more accurate actions more aligned with the expert data. Once expert actions have been retrieved we can speed up the sampling process in two distinct ways explained in item 1. (1) RAGDP-VP lets us skip the initial denoising stages and start later in the process depending on hyperparameter $r$. (2) RAGDP-VE instead simply reduces the amount of steps taken. We present the following contributions:

1. **RAGDP**: **R**etrive-**A**ugmented **G**enerative **D**iffusion **P**olicy, which allows for the storage and retrieval of expert trajectories from imitation learning data accompanied with two action sampling algorithms **RAGDP-VP** and **RAGDP-VE** which combined speeds up diffusion policy while keeping a high accuracy.

2. We provide extensive experimental results where we compare our work with current state-of-the-art methods and show that we can reduce the generation time while maintaining accuracy.

3. We demonstrate that RAGDP can be combined with existing speed-up methods, further improving their accuracy.

## 2 RELATED WORK

**Fast sampling methods for Diffusion Models**
Several approaches have been proposed to accelerate Diffusion Models. For example, Denoising Diffusion Implicit Models (DDIM) (Song et al., 2022) is a method that improves the sampling method of DDPM and can be adapted to trained models. Progressive Distillation (Salimans & Ho, 2022) is proposed as a method that uses knowledge distillation to reduce the sampling steps of the teacher model. Consistency Model (Song et al., 2023; Kim et al., 2024) is based on the Score-based Generative Models (Song et al., 2020), which is formulated as a stochastic differential equation for the Diffusion Models. Among these methods, since Progressive Distillation has a relatively high

learning cost, we design a model based on DDIM and the Consistency Models. In this paper, we show how these existing methods can be enhanced with RAGDP to improve the sampling efficiency further during the inference.

**Retrieval-Augmented Methods**

While the most widely application of RAG (Lewis et al., 2021) is Large Language Models (Naveed et al., 2024), there are several methods leveraging data retrieval in Diffusion Models. For instance, ReDi (Zhang et al., 2023) achieves speed-up by retrieving noisy data paired with data that has some noise removed. The Retrieval-Augmented Diffusion Model (Blattmann et al., 2022) retrieves both during training and inference, with retrieval during training to augment the data and retrieval during inference to search for appropriate conditional input.

In the context of robot learning, There are two lines of work leveraging trajectory retrieval. The first one is methods that retrieve trajectories during model training. Some methods aim to improve the dataset by retrieving data that is close to the expert during training. Nasiriany et al. (2022); Du et al. (2023) utilize Varial Auto Encoders (VAEs) (Kingma & Welling, 2022) to embed observations and the corresponding actions, and the trajectory retrievals are performed within the embedding space. The other is to retrieve during both training and inference; ReMoDiffuse (Zou et al., 2024) proposes to create a database of Text-Motion pairs and to make major architectural modifications to input the retrieved data into the model. READ (Oba et al., 2024) proposes a model that works with Image-Motion pairs, which retrieves trajectories during training and the image during inference.

However, these models primarily focus on refining trajectories or improving text and image generation, lacking application in action policy generation for imitation learning. RAGDP fills this gap by integrating retrieval-based expert trajectories into a diffusion policy framework for efficient action policy generation. By retrieving relevant expert demonstrations, RAGDP accelerates policy learning while maintaining high-quality action generation, making it particularly effective for real-time robotic tasks.

**Diffusion Model-based Data Editing**

Diffusion Models are powerful tools for image editing tasks, and their methods can be classified into three categories: training-based, testing-time finetuning, and training & finetuning free. While most methods target image and text modalities, SDEdit (Meng et al., 2022) is a method that can be utilized in the action space. SDEdit can obtain output in line with the input by reverse diffusion process from the input data with noise added at a specific step. Other methods that potentially can be applied in the action space include InstructPix2Pix (Brooks et al., 2023) and Denoising Diffusion Bridge Models (DDBM) (Zhou et al., 2023); however, these methods are not suited to improving the speed of generation. In this study, we focus on Training & Finetuning Free to consider methods that deal with the action space (Huang et al., 2024). This let us use RAGDP without any additional training of the diffusion model.

## 3 PRELIMINARIES

### 3.1 SCORE-BASED GENERATIVE MODELING

Score-Based Generative Model generalizes the Diffusion Model as a stochastic differential equation. Let $t$ denote the time direction and $\tau$ denote the diffusion direction; let $A_t$ be the trajectory at time step $t$ and $O_t$ be the data of the observed environment at that time. In the Diffusion Model, the direction in which the amount of noise increases is considered the forward process, while the direction in which the amount of noise decreases is considered the reverse process. Let $\sigma(\tau)$ be the sampling scheduler of the diffusion model and take the range of $\sigma \in [\sigma_{\min}, \sigma_{\max}]$. Then, we define two functions $F : \mathbb{R}^{D_A} \times [\sigma_{\min}, \sigma_{\max}] \to \mathbb{R}^{D_A}$ and $G : [\sigma_{\min}, \sigma_{\max}] \to \mathbb{R}$. Where $D_A$ is the dimension size of $A_t$. At this point, the forward process is as follows (Song et al. (2020)):

$$dA_t(\tau) = F(A_t(\tau), \sigma(\tau))d\sigma + G(\sigma(\tau))d\omega. \tag{1}$$

On the other hand, Reverse porcess is as follows:

$$dA_t(\tau) = \left[F(A_t(\tau), \sigma(\tau)) - \frac{1}{2}G(\sigma(\tau)^2\nabla_{A_t}\log p_\sigma(A_t(\tau)|O_t)\right]d\sigma + G(\sigma(\tau))d\omega. \tag{2}$$

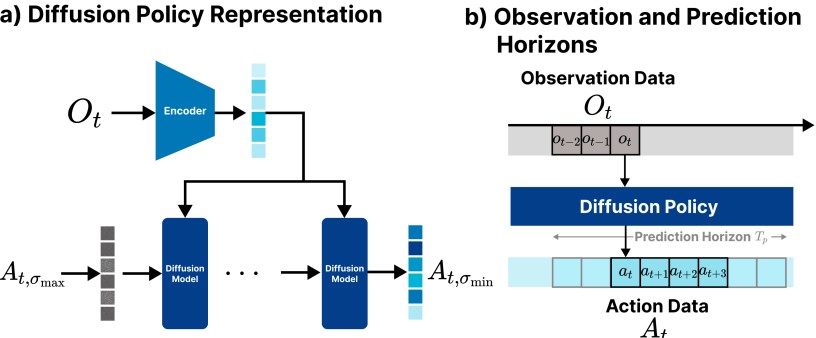

Figure 2: **a) Diffusion Policy Representation:** Diffusion Policy behaves as a diffusion model that takes data observed from the environment as conditional input and outputs trajectory data.
**b) Observation and Prediction Horizons:** The conditional input is $O_t$, chunked by $T_o$ steps of observed data $o_t$, and the generated behavior is $A_t$, chunked by $T_p$ steps of action step $a_t$.

The distribution that trajectory $A_t$ follows is a conditional probability distribution based on the observed data $O_t$.

**Variance Preserving Stochastic Differential Equations**
In the equation above, when $\sigma(\tau) = \tau$ and the functions are $F(A_t(\tau), \tau) = -\frac{1}{2}\beta(\tau)A_t(\tau)$ and $G(\tau) = \sqrt{\beta(\tau)}$, then the equation represents Variance Preserving Stochastic Differential Equations (VP-SDE). When the two functions are applied to the Equation 1 and the differential equation is solved, the general solution is as follows:

$$A_t(\tau) = \alpha(\tau)A_t + \sigma(\tau)z \quad \text{where} \quad z \sim \mathcal{N}(0, \mathbf{I}). \tag{3}$$

Where $\alpha(\tau)$ and $\sigma(\tau)$ are functions computed from $\beta(\tau)$ and have properties such as $\alpha(\tau)^2 + \sigma(\tau)^2 = 1$. Therefore, in VP-SDE, noise and data are mixed as a ratio at each step $\tau$, resulting in $\sigma(\tau) \in [0, 1]$. DDPM is classified as this type of Diffusion Models.

**Variance Exploding Stochastic Differential Equations**
Then, if the function is set $F(A_t(\tau), \tau) = 0$ and $G(\tau) = \sqrt{2\sigma(\tau)}$, called Variance Exploding Stochastic Differential Equations (VE-SDE). The general solution in this case is as follows:

$$A_t(\tau) = A_t + \sigma(\tau)z \quad \text{where} \quad z \sim \mathcal{N}(0, \mathbf{I}). \tag{4}$$

In VE-SDE, there is no limit to the amount of noise, and $\sigma \in [\sigma_{\min}, \sigma_{\max}]$. VE-SDE based EDM (Karras et al., 2022) models were employed in our experiments.

### 3.2 Diffusion models in Behavior Cloning

This section describes Diffusion Policy, a method of Behavior Cloning using the Diffusion Model. Let $o_t$ be the observed data at a certain time and $a_t$ be the behavior taken at that time, and $\mathcal{D} = \{(o_0^{(i)}, a_0^{(i)}), (o_1^{(i)}, a_1^{(i)}), ..., (o_{\mathcal{T}}^{(i)}, a_{\mathcal{T}}^{(i)})\}_{i=1}^N$ be the training data of the model, where $N$ is the number of episodes collected by the expert. The behavior of the Diffusion Policy is illustrated in Figure 2.

In the Diffusion Policy, the model takes observation data as input and outputs behavioral action data. The input observation data is chunked for the past $T_o$ steps $O_t = [o_t, o_{t-1}, ...]$. The output action data is chunked for $T_p$ steps of action step $a_t$ and is $A_t$. Only $T_a$ steps of it are executed. To generate $A_t$ using the Diffusion Model, Equation 2 can be utilized. $\nabla_{A_t} \log p_\sigma(A_t(\tau)|O_t)$ in Equation 2 is called the score function and is the quantity that the model should acquire in training $s_\theta = \nabla_{A_t} \log p_\theta(A_{t,\tau}|O_t)$. The optimization algorithm for learning is called score matching and is expressed by the following equation:

$$\mathcal{L}(\theta) = \mathbb{E}_{t \sim \mathcal{U}[0,\mathcal{T}], \tau \sim \mathcal{U}[0,T], A_t \sim p(A_t|O_t)} \left[ |s_\theta(A_{t,\tau}, \sigma_\tau, O_t) - \nabla_{A_t} \log p_{\sigma_\tau}(A_{t,\tau}|O_t)|^2 \right]. \tag{5}$$

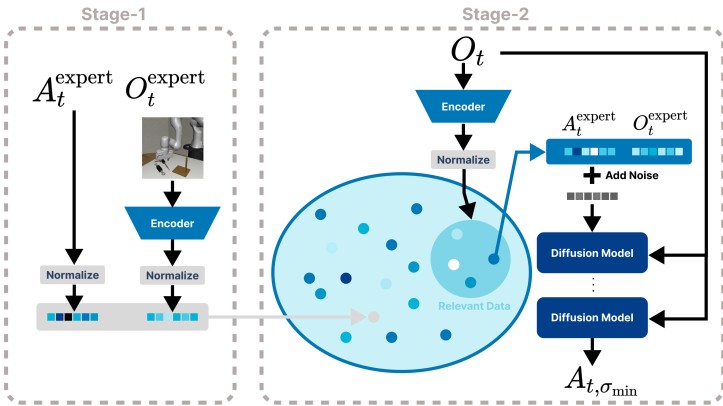

Figure 3: **RAGDP Representation:** The RAGDP is divided into two parts: the first is the knowledge base part, which is implemented as a vector database of observation and trajectory data pairs (section 4.1); the second is the diffusion model part, which outputs the final trajectory data via Diffusion Policy (section 4.2). The operation has two steps. Stage-1, which encodes the expert's data into a 1D vector and stores it in a vector database; Stage-2 consists of searching for relevant actions with observations made in the inference environment and generates them using a trained diffusion model.

## 4 APPLYING RETRIEVE-AUGMENTED GENERATION FOR DIFFUSION POLICY

This section describes the proposed method, RAGDP; the method consists of the retrieval part of the actions from the training dataset (section 4.1) and the action generation part using the retrieval action as an initial value for the denoising steps (section 4.2). The overall diagram of the proposed method is shown in Figure 3.

### 4.1 RETRIEVAL

**Implementing Vector Database**
The knowledge-base in RAGDP is a vector database consisting of pairs of observation data and corresponding expert trajectory data. When performing a search, the key vector is the observed data of the training data, and the value vector retrieved during the search corresponds to the trajectory data $A_t$ of the training data. The query vector during retrieval is the observed data during inference. In most of the previous studies, the embedding space is created by VAE for retrieval on observed data before retrieval. In this study, the vectors obtained from the encoders of the observed data implemented in Diffusion Policy are stored in the knowledge-base; the encoders in Diffusion Policy are CNN models in the case of images and identity functions in the case of states data. It has been shown that this encoder is better trained simultaneously with Diffusion Policy than pre-trained alone on a large data set.

The vector database is implemented in Facebook AI Similarity Search (FAISS) (Johnson et al., 2017), which uses a search method that indexes from L2 distances in Euclidean space.

**Retrieving Strategy**
In searching the vector database, the following issues are considered: when to search for a time step $t$, how many samples with the highest search similarity should be obtained, and whether to use a threshold for the search. RAGDP simply searches at every time step, does not use a threshold when searching, and works to obtain the top one similarity sample.

### 4.2 GENERATION

Next, we will explain how to generate the final sample based on the samples obtained from the Knowledge-base. The proposed method is based on SDEdit as a method that can be adapted to the behavioral space from two perspectives: it can be used with existing Diffusion Models and it

---

**Algorithm 1** RAGDP-VP (DDPM) Sampling Algorithm

---

**Require:** diffuse rate $r$, total denosing steps $T$, denosing scheduler $\sigma_\tau$, total episode steps $\mathcal{T}$, pre-trained model parameters $\theta$, vector database $\{(O_i^{\text{exp}}, A_i^{\text{exp}}) | i \in \{1, 2, \ldots, N_{\text{data}}\}\}$.

1: **for** $t = 1$ to $\mathcal{T}$ **do**
2:      Observe $O_t$
3:      $i \leftarrow \underset{n=1\ldots N_{\text{data}}}{\text{argmin}} \|O_t - O_n^{\text{exp}}\|$
4:      $A^{\text{ret}} \leftarrow A_i^{\text{exp}}$
5:      $\tau^* \leftarrow \lfloor (1 - r)T \rfloor$
6:      $z \sim \mathcal{N}(0, \mathbf{I})$
7:      $A_{t,\tau^*} \leftarrow \sqrt{\overline{\alpha}_{\tau^*}} A^{\text{ret}} + \sqrt{1 - \overline{\alpha}_{\tau^*}} z$
8:      **for** $\tau = \tau^*$ to $0$ **do**
9:          $z \sim \mathcal{N}(0, \mathbf{I})$ **if** $\tau > 0$ **else** $z = 0$
10:          $A_{t,\tau-1} = \frac{1}{\sqrt{\alpha_\tau}} \left( A_{t,\tau} - \frac{1-\alpha_\tau}{\sqrt{1-\overline{\alpha}_\tau}} z_\theta(A_{t,\tau}, \tau, O_t) \right) + \sigma_\tau z$
11:      **end for**
12:      Execute $A_{t,0}$
13: **end for**

---

can speed up the sample speed. The following two methods were implemented for the generation algorithm.

**RAGDP-VP**

In VP-SDE, the parameters in Equation 3 are constrained by $\alpha(\tau)^2 + \sigma(\tau)^2 = 1$. This means that the magnitude of the noise and the action are determined by a ratio. Therefore, the action retrieved from the database is used to calculate the final output from the ratio of action and noise corresponding to the starting diffusion step $\tau_0$. RAGDP-VP introduces a hyperparameter $r$, which determines the initial position to start the denoising process. If the number of diffusion steps is $T$ and the step to start generating is $\tau_0$, then $r = \tau_0/T$. Since the parameter takes the range $0 < r < 1$, the number of steps to generate samples is $(1 - r)T$, which enables faster processing. In principle, RAGDP-VP can be applied to both VP-SDE and VE-SDE Diffusion Models and Consistency Models. The DDPM-based RAGDP-VP is shown in Algorithm 1.

**RAGDP-VE**

In the case of VE-SDE, $\alpha(\tau)$ in Equation 3 is fixed by $\alpha(\tau) = 1$. Therefore, there is no limit on the size of the action and noise. Therefore, the action taken from the database adds noise of a magnitude corresponding to the starting diffusion step $\tau_0$, and the output is obtained where this noise becomes smaller. Therefore, RAGDP-VE always adds $\sigma_{\max}$ without changing the amount of initial noise and only changes the number of sample steps. Similarly, a hyperparameter $r$ is introduced, which similarly generates samples by calculating $(1 - r)T$ steps. RAGDP-VE can be applicable for VE-SDE based Diffusion Model only and Consistency Models. The EDM-based RAGDP-VE is shown in Algorithm 2.

## 5 EXPERIMENTS

In this section, we evaluate the effectiveness of RAGDP in enhancing the performance of traditional Diffusion Policy. The performance is benchmarked on the Behavior Cloning datasets: Robomimic (Mandlekar et al., 2021) and Push-T. Our implementation of the diffusion model integrates various diffusion policies, including DDPM, DDIM, and EDM. Additionally, we incorporate Consistency Policy based on Consistency Trajectory Models (Prasad et al., 2024) for comparative analysis. We aim to investigate the following: (1) How the performance of RAGDP varies as the number of steps changes. (2) The impact of the hyperparameter $r$ on both inference speed and accuracy. (3) We make a comparison of RAGDP-VP and RAGDP-VE.

### 5.1 EVALUATION SETUP

We trained the Diffusion Policy models using the Behavior Cloning dataset, as in the Equation 5, and then creating a vector database from the training dataset. Finally, the accuracy and generation

**Algorithm 2** RAGDP-VE (EDM) Sampling Algorithm

**Require:** diffuse rate $r$, total denosing steps $T$, denosing scheduler $\sigma_\tau$, total episode steps $\mathcal{T}$, pre-trained model parameters $\theta$, vector database $\{(O_i^{\text{exp}}, A_i^{\text{exp}})|i \in \{1, 2, \ldots, N_{\text{data}}\}\}$.

1: **for** $t = 1$ to $\mathcal{T}$ **do**
2:      Observe $O_t$
3:      $i \leftarrow \underset{n=1\ldots N_{\text{data}}}{\text{argmin}} \|O_t - O_n^{\text{exp}}\|$
4:      $A^{\text{ret}} \leftarrow A_i^{\text{exp}}$
5:      $n \leftarrow (1 - r)T$
6:      $\Delta\tau \leftarrow \lfloor \frac{T}{n} \rfloor$
7:      $A_{t,T} \leftarrow A^{\text{ret}} + \sigma_{\max}z$
8:      $\tau \leftarrow T$
9:      **for** $j = 1$ to $n$ **do**
10:         $A_{t,\tau-\Delta\tau} = A_{t,\tau} + (\sigma_\tau^2 - \sigma_{\tau-\Delta\tau}^2)s_\theta(A_{t,\tau}, \sigma_\tau, O_t)$
11:         $\tau \leftarrow \tau - \Delta\tau$
12:      **end for**
13:      Execute $A_{t,0}$
14: **end for**

Table 1: **Task Details.** #Rob: number of robots, #Obj: number of objects, ActD: action dimension, PH: proficient-human demonstration, MH: multi-human demonstration, Steps: max number of rollout steps.

| Task | #Rob | #Obj | ActD | #PH | #MH | Steps |
|---|---|---|---|---|---|---|
| Square | 1 | 1 | 7 | 200 | 300 | 400 |
| ToolHang | 1 | 2 | 7 | 200 | - | 700 |
| Transport | 2 | 3 | 14 | 200 | 300 | 700 |
| Push-T | 1 | 1 | 2 | 200 | - | 300 |

speed of the trained model were measured in a test environment. Table 1 shows the tasks selected for evaluation.

The tasks were benchmarked in a simulation environment, covering SQUARE-PH, SQUARE-MH, TOOLHANG-PH, and PUSH-T. The performance of each task is the average of the models trained on 3 different seeds. 56 different seeds were available in the evaluation environment, for a total of 168 measurements per task. The evaluation seed was not included in the training seed. For all experiments, state data was used for observations. The column "Steps" in Table 1 specifies the maximum number of steps allowed per episode. For SQUARE-PH, SQUARE-MH, and TOOLHANG-PH, accuracy is reported as the average success rate of the task. For PUSH-T the accuracy measures the target area coverage. The task was also validated in TRANSPORT-PH as a task with a large action dimension $D_A$. Here, SQUARE-PH, MH and PUSH-T are single-step tasks, while TOOLHANG and TRANSPORT-PH are multi-step tasks as they move multiple objects.

Table 2: **Model Details.** The number of sample steps of the model used in the experiment and the method of action generation are shown in the table.

| Models | Sampling Steps $T$ | Diffusion Policy | RAGDP-VP | RAGDP-VE |
|---|---|---|---|---|
| DDPM | 100 | ✓ | ✓ | ✗ |
| DDIM | 25 | ✓ | ✓ | ✗ |
| EDM | 40 | ✓ | ✓ | ✓ |
| CTM | 4 | ✓ | ✓ | ✓ |

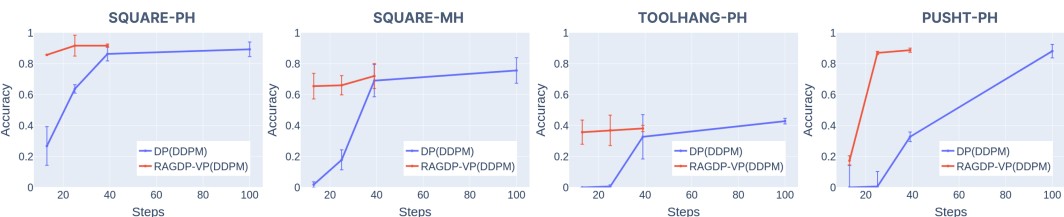

Figure 4: **Results of utilizing Diffusion Policy and RAGDP in the DDPM model.** Using RAGDP-VP sampling when utilizing RAGDP in the DDPM model. DDPM tends to rapidly lose accuracy when de-noising steps are reduced, but the use of RAGDP-VP shows that accuracy is robustly maintained, except for PUSHT.

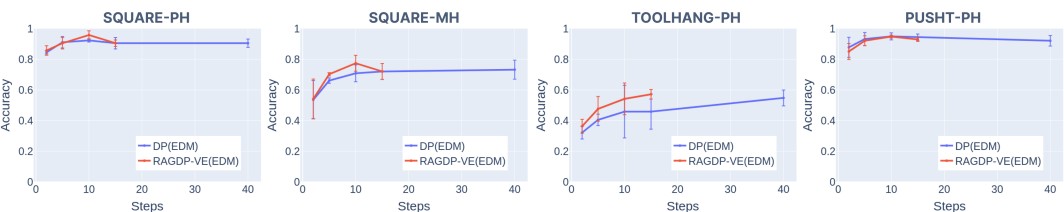

Figure 5: **Results of utilizing Diffusion Policy and RAGDP in the EDM model.** Using RAGDP-VE sampling when utilizing RAGDP in the EDM model. EDM models show a relatively gradual decrease in accuracy with respect to denoising step reductions. However, by utilizing RAGDP-VE, the accuracy of the same models was improved.

We benchmarked the following diffusion-based models: DDPM, DDIM, and EDM for Diffusion Policy (DP), and Consistency Policy (CP) models based on CTM. DDIM can be combined with RAGDP-VP as there is a DiffEdit (Couairon et al., 2022) selection study. The DDPM, DDIM, EDM, and CTM models were used for RAGDP-VP, while only EDM and CTM were used for RAGDP-VE. The CTM is trained by knowledge-distillation of the trained EDM as a teacher model. DPM-Solver++ (Lu et al., 2023) is used for EDM sampling.

## 5.2 EFFECTIVENESS OF RAGDP

We compared the effectiveness of the proposed method against existing Diffusion Policies, specifically those implemented with the DDPM and EDM models. We then demonstrated the relationship between accuracy and the number of sampling steps with and without adaptation of RAGDP. The results of the comparison for each model are shown in Figure 4 and Figure 5. The numbers shown in the figure represent the average of the 3 seeds.

The figures demonstrate that, for RAGDP-VP(DDPM), accuracy remains relatively stable even as the number of sampling steps decreases except for PUSHT which is more challenging for fewer steps for both DDPM models. For RAGDP-VE(EDM), there is slight decline but still obtaining a higher accuracy than EDM only.

In the Appendix C, we show graphs of RAGDP adapted to DDIM and Consistency Models, which is a method for speeding up Diffusion Models, affected by the trade-off relationship in RAGDP-VP. Conversely, the use of RAGDP in Consistency Models can extend the performance of existing methods.

## 5.3 SPEEDUPS ON DIFFUSION POLICY

Sampling speed and accuracy measurements were then performed on various models. The experimental accuracy results are shown in Table 3 for RAGDP hyperparameters $r$, with values of $r = 0.25, 0.50$, and $0.75$. The combined speed and accuracy results are also shown in Figure 6, indicating that the RAGDP can be used to increase sample speed without compromising accuracy.

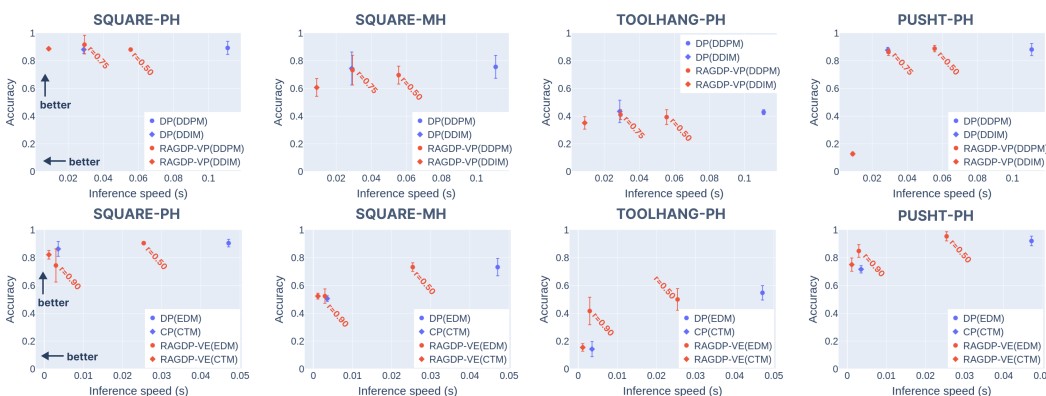

Figure 6: **Inference speed and accuracy:** 3 seeds average rewards are shown as accuracy on the vertical axis and sampling speed on the horizontal axis. Speeds were measured on RTX 3060. The upper left direction of the graph indicates better performance. The results of RAGDP-VP of DDPM are compared with DDIM. The upper half of the figure shows that the accuracy of RAGDP-VP is comparable to DDIM. As RAGDP-VP can also be adapted to DDIM, this result is included in the Appendix C. Next, the lower half of the figure shows a comparison of RAGDP-VE with CTM. The results show that by utilising RAGDP-VE for EDM, the accuracy reaches the same or better than that of CTM at the same speed as CTM. The more results of adapting RAGDP with CTM are shown in Figure 14.

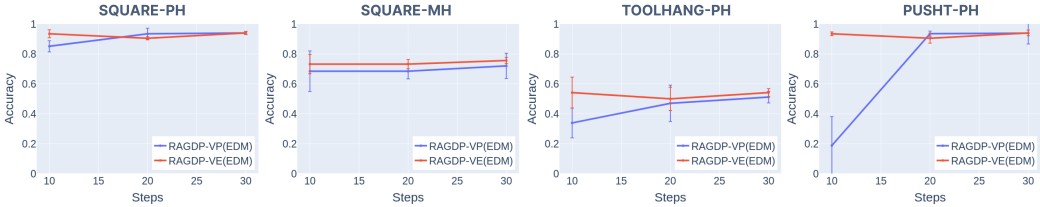

Figure 7: **Comparison of RAGDP-VP and RAGDP-VE performance in VE-SDE based Diffusion Model:** A comparison of VE-SDE-based EDM models in PUSHT-PH and TOOLHANG-PH shows that RAGDP-VP is less accurate with respect to sampling steps, while RAGDP-VE is more robust.

We do further investigations on how the average search distance affects the accuracy of RAGDP for the more challenging environments in the Appendix B.2.

## 5.4 RAGDP-VP vs. RAGDP-VE

RAGDP-VP can also be utilized for VE-SDE-based models. In fact, SDEdit has shown that it can be used in VE-SDE-based models to generate images according to the conditions. Therefore, it is necessary to investigate how it behaves in the action space, so we compared the accuracy of sampling with RAGDP-VP and RAGDP-VE in the VE-SDE-based EDM. The Figure 7 shows the results of comparing the change in accuracy for each sampling technique by reducing the number of sampling steps. RAGDP-VP has a trade-off between faithfulness to the input and realism. Therefore, when the parameter $r$ is large, the de-noising step is smaller and the amount of noise given is smaller, so realism tends to be weaker and less accurate.

The Appendix C shows comparative results in the case of CTM: for EDM, RAGDP-VE is better, but in some cases RAGDP-VP is better for CTM, where the noise magnitude can be controlled.

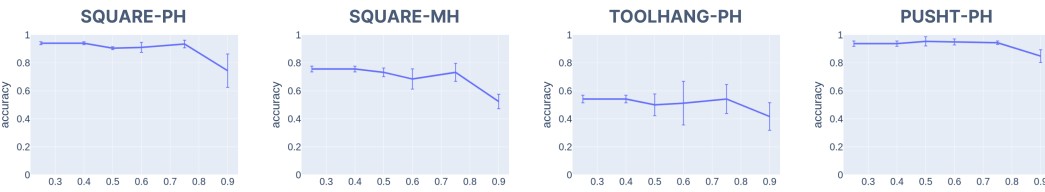

Figure 8: **Accuracy and hyperparameter $r$:** The effect of hyperparameter $r$ on accuracy of the EDM model is shown. It can be seen that the larger the $r$, the faster the generation speed increases, but the accuracy tends to decrease.

## 5.5 CHOICE OF HYPERPARAMETER

Consider the choice of hyperparameter $r$. In image-based diffusion models, there is a known trade-off between faithfulness to the input and realism. If the number of steps to denoise with respect to the input is small, realistic samples cannot be generated, and conversely, if the number of denoise steps with respect to the input is large, the faithfulness to the output to be obtained is reduced. Therefore, in the image-based case, the weightspot is chosen, which is expressed in $r$ as $r \in [0.4, 0.7]$. The proposed method also measured the hyper-parameters with the EDM model, as shown in Figure 8. From the figure, it can be seen that for each task, performance tends to decrease when $r = 0.75$ or higher. Therefore, it is considered optimal to determine parameters around this point where a trade-off between task accuracy and speed can be made.

## 6 CONCLUSION

In this study, speed and accuracy benchmarks were created by utilising Diffusion Policy in DDPM, EDM and CTM models. Then, using a vector database as a knowledge base, we proposed RAGDP, a method that can improve sample speed without requiring additional training and without reducing the accuracy of Diffusion Policy. in RAGDP, sample speed is determined by the parameter $r$ and can generate behaviour for VP-SDE and VE-SDE based Diffusion Models using the RAGDP-VP and RAGDP-VE sampling methods. The proposed method is shown to be robust to a reduced number of steps. And RAGDP-VP was shown to improve the performance of existing models by adjusting the noise magnitude when utilising DDPM and when utilising CTM; RAGDP-VE was shown to be effective for EDM-based models when speeding up the process.

However, several points need to be improved in the future. First, RAGDP is sensitive concerning existing models and hyperparameters. Therefore, it is necessary to select hyperparameters in line with specific tasks. This is because the initial values of the generation process vary. For example, as Align Your Steps (Sabour et al., 2024) improves accuracy by compensating for the discretization error of the scheduler, it may be necessary to consider a scheduler that follows the initial values. Second, this study does not discuss the creation of a semantic space when searching with observed data. Future work should investigate improving the embedded space to be searched for in behavioral data as well.

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

Table 3: **Accuracy results of single step tasks.** The table shows the results of accuracy when generating actions with the existing Diffusion Policy (DP), Consistency Policy (CP), and RAGDP, measuring the accuracy when the parameter $r$ of RAGDP is varied in three different patterns.

|  | SQUARE (PH) | SQUARE (MH) | PUSH-T |
|---|---|---|---|
| DP(DDPM) | $0.8929 \pm 0.0473$ | $0.7560 \pm 0.0826$ | $0.8811 \pm 0.0436$ |
| RAGDP-VP(DDPM) $r = 0.25$ | $0.8572 \pm 0.0358$ | $0.7203 \pm 0.0806$ | $0.8853 \pm 0.0248$ |
| RAGDP-VP(DDPM) $r = 0.50$ | $0.8810 \pm 0.0104$ | $0.6607 \pm 0.0619$ | $0.8881 \pm 0.0230$ |
| RAGDP-VP(DDPM) $r = 0.75$ | $0.9167 \pm 0.0676$ | $0.6548 \pm 0.0826$ | $0.8632 \pm 0.0249$ |
| DP(EDM) | $0.9048 \pm 0.0273$ | $0.7322 \pm 0.0619$ | $0.9205 \pm 0.0344$ |
| RAGDP-VE(EDM) $r = 0.25$ | $0.9405 \pm 0.0103$ | $0.7560 \pm 0.0207$ | $0.9372 \pm 0.0191$ |
| RAGDP-VE(EDM) $r = 0.50$ | $0.9048 \pm 0.0103$ | $0.7322 \pm 0.0310$ | $0.9537 \pm 0.0328$ |
| RAGDP-VE(EDM) $r = 0.75$ | $0.9346 \pm 0.0273$ | $0.7322 \pm 0.0644$ | $0.9438 \pm 0.0127$ |
| RAGDP-VE(EDM) $r = 0.90$ | $0.7441 \pm 0.1190$ | $0.5239 \pm 0.0516$ | $0.8485 \pm 0.0454$ |
| DP(DDIM) | $0.8810 \pm 0.0273$ | $0.7441 \pm 0.1190$ | $0.8769 \pm 0.0202$ |
| RAGDP-VP(DDIM) $r = 0.25$ | $0.8810 \pm 0.0273$ | $0.7441 \pm 0.0450$ | $0.8660 \pm 0.0161$ |
| RAGDP-VP(DDIM) $r = 0.50$ | $0.9108 \pm 0.0179$ | $0.7441 \pm 0.0413$ | $0.8173 \pm 0.0162$ |
| RAGDP-VP(DDIM) $r = 0.75$ | $0.8870 \pm 0.0104$ | $0.6667 \pm 0.0574$ | $0.1277 \pm 0.0127$ |
| CP(CTM) | $0.8393 \pm 0.0536$ | $0.6310 \pm 0.0207$ | $0.7985 \pm 0.0265$ |
| RAGDP-VP(CTM) $r = 0.25$ | $0.9048 \pm 0.0546$ | $0.5179 \pm 0.0619$ | $0.7996 \pm 0.0180$ |
| RAGDP-VP(CTM) $r = 0.50$ | $0.8989 \pm 0.0273$ | $0.5953 \pm 0.0826$ | $0.8077 \pm 0.0106$ |
| RAGDP-VP(CTM) $r = 0.75$ | $0.8155 \pm 0.0806$ | $0.5120 \pm 0.0722$ | $0.1548 \pm 0.0273$ |
| RAGDP-VE(CTM) $r = 0.25$ | $0.7738 \pm 0.0450$ | $0.5298 \pm 0.1341$ | $0.7629 \pm 0.0289$ |
| RAGDP-VE(CTM) $r = 0.50$ | $0.8096 \pm 0.0744$ | $0.5238 \pm 0.0207$ | $0.7215 \pm 0.0260$ |
| RAGDP-VE(CTM) $r = 0.75$ | $0.8215 \pm 0.0309$ | $0.5238 \pm 0.0207$ | $0.7503 \pm 0.0476$ |

Table 4: **Accuracy results of multi step tasks.** Accuracy of action generation in multi-step tasks.

|  | TOOLHANG (PH) | TRANSPORT (PH) |
|---|---|---|
| DP(DDPM) | $0.4286 \pm 0.0179$ | $0.7679 \pm 0.0179$ |
| RAGDP-VP(DDPM) $r = 0.25$ | $0.4643 \pm 0.0779$ | $0.7619 \pm 0.0372$ |
| RAGDP-VP(DDPM) $r = 0.50$ | $0.3929 \pm 0.0536$ | $0.7857 \pm 0.0644$ |
| RAGDP-VP(DDPM) $r = 0.75$ | $0.4107 \pm 0.0357$ | $0.7500 \pm 0.0618$ |
| DP(EDM) | $0.5477 \pm 0.0516$ | $0.7679 \pm 0.0715$ |
| RAGDP-VE(EDM) $r = 0.25$ | $0.5417 \pm 0.0273$ | $0.8155 \pm 0.0273$ |
| RAGDP-VE(EDM) $r = 0.50$ | $0.5000 \pm 0.0779$ | $0.7738 \pm 0.0273$ |
| RAGDP-VE(EDM) $r = 0.75$ | $0.5417 \pm 0.1032$ | $0.8095 \pm 0.0844$ |

# A    DETAILED EXPERIMENTAL RESULTS

Detailed results of measuring the accuracy of the 3-seed average with various hyperparameters are shown in the Table 3 and Table 4.

Figure 9 shows the effect of reducing the number of steps when utilising RAGDP-VP and RAGDP-VE in Transport-PH.

# B    MORE STUDIES ON KNOWLEDGE-BASE

## B.1    IMPLEMENTING KNOWLEDGE-BASE

In this section, we describe how we converted the training data to knowledge-base. The training data to be stored in knowledge-base is based on the policy of storing arrays similar to the Diffusion Policy.

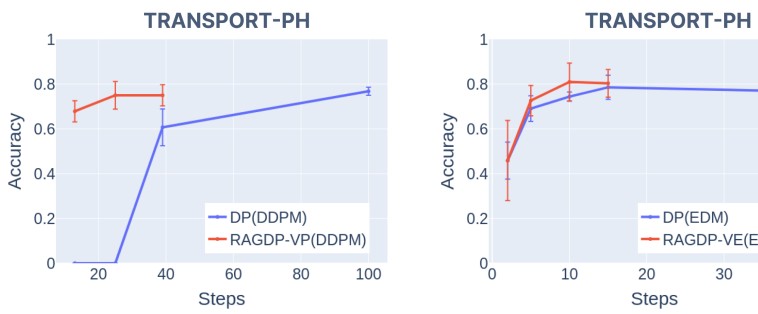

Figure 9: **Results of utilizing RAGDP-VP and RAGDP-VE in TRANSPORT-PH task:** It can be seen that RAGDP works even in the TRANSPORT task, where the action dimension is twice as large.

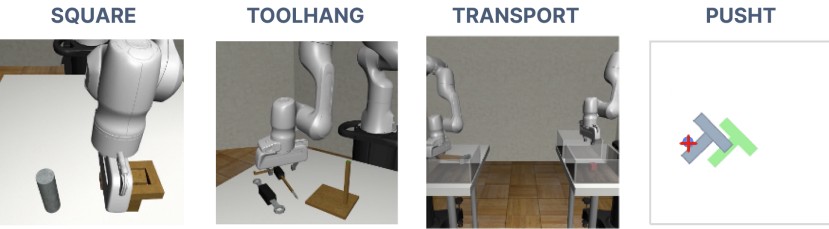

Figure 10: **Behavior Cloning tasks:** In this study, 3 datasets of single robot tasks, SQUARE, TOOLHANG, TRANSPORT and PUSHT, were used.

Table 5: **Row numbers of the Knowledge-base.**

| Tasks | Row numbers |
| --- | --- |
| SQUARE-PH | 28754 |
| SQUARE-MH | 78631 |
| TOOLHANG-PH | 94562 |
| TRANSPORT-PH | 92352 |
| PUSHT-PH | 24208 |

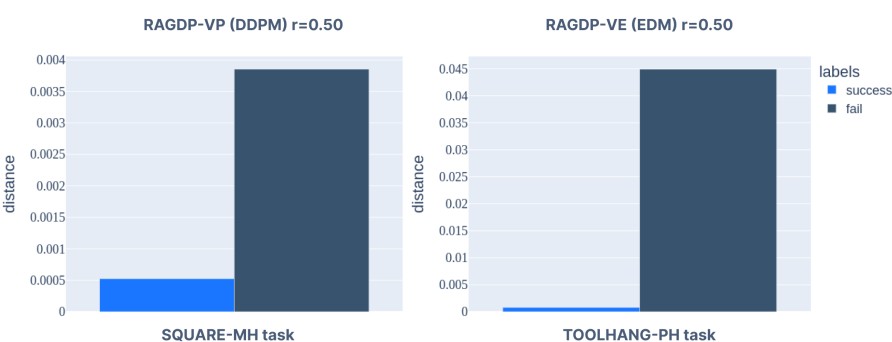

Figure 11: **Knowledge-base average search distance results.:** The results of inferring RAGDP-VP (DDPM) with the TOOLHANG-PH task and RAGDP-VE (EDM) with the SQUARE-MH task are shown.

The retrieved vector is an array of $T_o \times D_o$, and the output vector is $T_A \times D_A$. Table 5 shows the number of rows in the knowledge-base created for all training data. All data is normalized prior to input, as is the training data.

## B.2 KNOWLEDGE-BASE STUDIES

From the above experimental results, we obtained that there are some patterns in which the use of RAGDP improves accuracy slightly. Therefore, it is necessary to investigate how the use of Knowledge-base affects accuracy. As a simple experiment, we recorded the similarity of searches for a task and examined the patterns of success and failure. The tasks selected were TOOLHANG-PH and SQUARE-MH, which have relatively high failure patterns. The model took the average of the three seeds of RAGDP-VP (DDPM) and RAGDP-VE (EDM). The results are shown in Figure 11. The vertical axis of the figure represents the average of the similarity distance $L = \frac{1}{\mathcal{T}} \sum_{t=1}^{\mathcal{T}} |O_t - O_t^{\mathrm{expert}}|^2$ obtained for all episodes and test environments. From the figure, it can be seen that the more successful the task is, the smaller the distance obtained from the knowledge-base.

In the Appendix B.3, we also experimented with the behavior of accuracy when the knowledge-base was created with untrained data. From the experimental results, it was found that the accuracy of the knowledge-base was not degraded when it was created with trained data.

## B.3 KNOWLEDGE-BASE AND TRAINING DATASET

In robomimic, we also experimented with the behavior of the knowledge-base when it is created with untrained data: the PH and MH datasets. PH is the data that performed skilled human. The MH task consists of "better", "okay", and "worst" data. Therefore, as an experiment, we created a knowledge-base in SQUARE-MH for the model trained in SQUARE-PH and a knowledge-base for the model trained in SQUARE-MH, and measured the accuracy of each when generated by RAGDP. The results are shown in Table 6 and Table 7, respectively.

## B.4 STUDY OF KNOWLEDGE-BASE SIZE

Variation with respect to the amount of databases was investigated: as the number of datasets in behavior cloning is limited, experiments were conducted with a small amount of databases. In the

Table 6: **Result of creating a knowledge-base with SQUARE-MH for a model trained with SQUARE-PH.** Each accuracy represents the average of three seeds. The model is RAGDP-VE (EDM).

| Knowledge-base | Accuracy ($r = 0.25$) | Accuracy ($r = 0.50$) | Accuracy ($r = 0.75$) |
|---|---|---|---|
| SQUARE-PH (base) | $0.9405 \pm 0.0103$ | $0.9048 \pm 0.0103$ | $0.9346 \pm 0.0273$ |
| SQUARE-MH | $0.8870 \pm 0.0450$ | $0.8869 \pm 0.0744$ | $0.7917 \pm 0.1077$ |

Table 7: **Result of creating a knowledge-base with SQUARE-PH for a model trained with SQUARE-MH.** Each accuracy represents the average of three seeds. The model is RAGDP-VE (EDM).

| Knowledge-base | Accuracy ($r = 0.25$) | Accuracy ($r = 0.50$) | Accuracy ($r = 0.75$) |
|---|---|---|---|
| SQUARE-MH (base) | $0.7560 \pm 0.0207$ | $0.7322 \pm 0.0310$ | $0.7322 \pm 0.0644$ |
| SQUARE-PH | $0.6965 \pm 0.0309$ | $0.7262 \pm 0.0677$ | $0.7143 \pm 0.0357$ |

experiment, 100% of the database was created with all training data, and the accuracy and search distance were investigated when the database was varied to 10% and 1%. The results are shown in Figure 12. From the figure, it can be seen that the retrieval distance tends to decrease as the database size increases. However, it can be seen that accuracy has not changed significantly. Therefore, accuracy is not considered to be directly related to retrieve distance. Therefore, it is thought that quality, not quantity, may be important in terms of the data that should be stored in the database. Research (Du et al., 2023) has shown that accuracy can be improved with less data by retrieving data at the time of training.

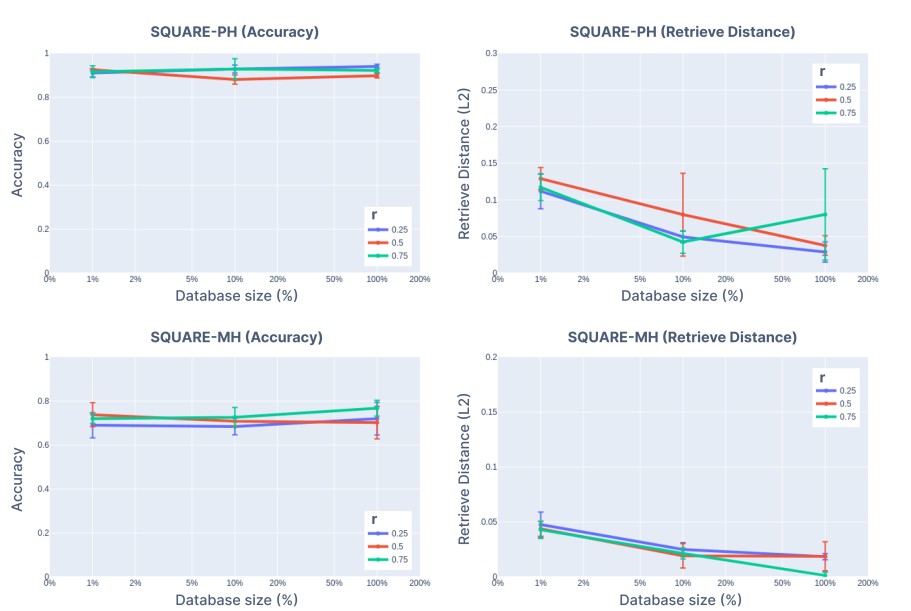

Figure 12: **Result of varying the size of the database.** The behavior of SQUARE-PH and MH with respect to the respective hyperparameter $r$ was measured with the EDM-based RAGDP-VE. The left-hand side shows the results for accuracy and database size, while the right-hand side shows the results for retrieve distance and database size. The results show that the retrieve distance tends to decrease as the database size increases. However, the accuracy has not changed significantly.

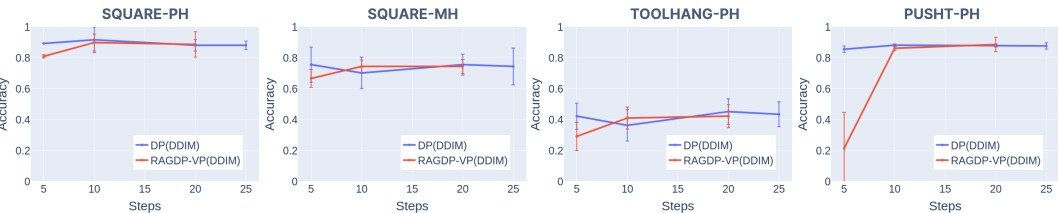

Figure 13: **Results of utilizing Diffusion Policy and RAGDP in the DDIM model.** Using RAGDP-VP sampling when utilizing RAGDP in the DDIM model. RAGDP-VP has a trade-off between faithfulness for the input and realism for the output; in the case of DDIM, the small number of sampling steps shows that the trade-off effect is stronger when the number of steps is smaller.

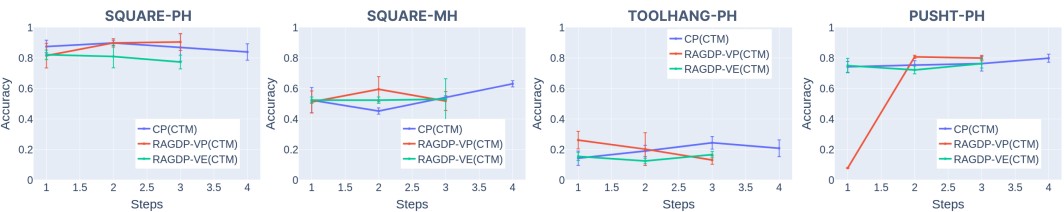

Figure 14: **Results of utilizing Consistency Policy and RAGDP in the CTM model.** Both RAGDP-VP and RAGDP-VE can be used in CTM. The results show that when the number of steps is 2, RAGDP-VP reaches the same or better accuracy than the existing CTM. This is because when using RAGDP in CTM, if the amount of noise is too large for the action taken in the search, the effect will be small. Therefore, it is necessary to determine the appropriate noise level. Conversely, RAGDP-VE cannot control the size of the noise, which means that it is equal to or less than existing CTMs.

## C  RAGDP IN FAST SAMPLING METHODS

This section shows the results of adapting RAGDP in existing acceleration methods. First, the results of adapting RAGDP-VP to DDIM are shown in  Figure 13.

The results of adapting the RAGDP to the CTM are then shown in the Fig  Figure 14. Although this study experiments with methods that focus on reducing the number of steps, it can be said that, in the case of CTM, the behaviour in the amount of noise is also important.

## D  CODE IMPLEMENTATIONS

```python
class RobotFAISS(object):
    def __init__(
            self,
            index_name:str, # toolhang.index
            vector_dimensions:int,
            vector_db_folder:str='./db',
        ) -> None:
        self.index_name = index_name
        self.dict_name = index_name.replace(".index", ".pkl")
        self.vector_dimensions = vector_dimensions
        self.vector_db_folder = vector_db_folder

        self.index : Optional[faiss.IndexFlatL2] = None
        self.vector_dict : Dict[int, np.array] = {}

    def initialize_index(self) -> None:
        index = faiss.IndexFlatL2(self.vector_dimensions)
        index_path = os.path.join(self.vector_db_folder, self.index_name)
```

```python
        write_index(index, index_path)
        self.index = index

    def initialize_dict(self) -> None:
        vector_dict : Dict[int, np.array] = {}
        dict_path = os.path.join(self.vector_db_folder, self.dict_name)
        with open(dict_path, 'wb') as f:
            pickle.dump(vector_dict, f)
        self.vector_dict = vector_dict

    def load_index(self) -> faiss.IndexFlatL2:
        index_path = os.path.join(self.vector_db_folder, self.index_name)
        index = read_index(index_path)
        return index

    def load_dict(self) -> Dict[int, np.array]:
        dict_path = os.path.join(self.vector_db_folder, self.dict_name)
        with open(dict_path, "rb") as f:
            vector_dict = pickle.load(f)
        return vector_dict

    def load(self):
        self.index = self.load_index()
        self.vector_dict = self.load_dict()

    def initialize_db(self, input_vectors:List[np.array], result_vectors:
    List[np.array]):
        """
            - input_vectors: obs_vectors reshaped in 1D (normalized)
            - result_vectors: action_vectors reshaped in 1D (noramlized)
        """
        self.initialize_index()
        self.initialize_dict()

        # Create Dict
        for i, (input_vector, result_vector) in enumerate(zip(
    input_vectors, result_vectors)):
            self.vector_dict[i] = result_vector
        dict_path = os.path.join(self.vector_db_folder, self.dict_name)
        with open(dict_path, 'wb') as f:
            pickle.dump(self.vector_dict, f)

        # Create Index
        vectors = np.array(input_vectors, dtype=np.float32)
        index_path = os.path.join(self.vector_db_folder, self.index_name)
        self.index.add(vectors)
        write_index(self.index, index_path)

    def search(self, query_vector:np.array, k:int) -> List[np.array]:
        query_vector = query_vector.reshape(1, -1)
        scores, indices = self.index.search(query_vector, k)
        result_vectors = [
            self.vector_dict[int(i)] for i in indices[0]
        ]
        return result_vectors
```

Listing 1: FAISS Vector Database code

