# OpenReview forum: "RAGDP: Retrieve-Augmented Generative Diffusion Policy"
_ICLR.cc/2025/Conference — Submitted to ICLR 2025_

### Official Review · Reviewer_q22q · 2024-10-27

**Soundness:** 2
**Presentation:** 2
**Contribution:** 1
**Rating:** 3
**Confidence:** 3

**Summary:**

The paper proposes RAGDP, a new method aimed at enhancing the efficiency of diffusion policies in imitation learning by leveraging Retrieval-Augmented Generation (RAG). Traditional diffusion policies require multiple denoising steps, resulting in long generation times, which methods like DDIM and Consistency Models attempt to address at the expense of accuracy. RAGDP utilizes a vector database to store and retrieve pairs of expert observation and action data based on encoded embeddings, enabling faster action generation tailored to the current environment. The authors introduce two specific algorithms, RAGDP-VP and RAGDP-VE, corresponding to different types of diffusion models. Experimental results demonstrate that RAGDP significantly reduces generation time while maintaining accuracy and can be effectively integrated with existing acceleration methods.

**Strengths:**

#### Originality
The paper presents a novel approach called RAGDP (Retrieve-Augmented Generative Diffusion Policy) that significantly enhances the efficiency of diffusion policies in imitation learning. By leveraging Retrieval-Augmented Generation (RAG), the authors introduce a fresh and creative solution to the problem of long generation times associated with traditional diffusion policies. The proposal of two distinct algorithms, RAGDP-VP and RAGDP-VE, further underscores the paper's originality by catering to different diffusion model types.

#### Quality
The paper demonstrates high-quality research through extensive experimentation and empirical validation. The authors provide comparative analyses with existing methods like DDIM and Consistency Models, showcasing the effectiveness of RAGDP in enhancing speed without sacrificing accuracy.

#### Clarity
The paper is well-written and clearly structured, making it accessible to readers with a background in imitation learning and diffusion models.

#### Significance
The significance of the proposed RAGDP method is evident in its potential to drastically reduce the computational cost of diffusion policies in imitation learning tasks. By maintaining high accuracy while improving speed, RAGDP addresses a critical bottleneck in practical applications, especially in real-time robotic tasks.

**Weaknesses:**

- **Limited Novelty in Retrieval-Augmented Techniques:** While RAGDP is creative in combining retrieval-augmented generation with diffusion policies, the concept of retrieval-augmented generation itself is not entirely new. Prior works like RAG (Lewis et al., 2021) for text generation and ReDi (Zhang et al., 2023) for image generation are mentioned in the paper. The novelty mainly lies in its application to diffusion policies for imitation learning, which may not be sufficient to stand out unless significantly differentiated from these prior works.

- **Hyperparameter Sensitivity:** The experiments indicate that RAGDP's performance is sensitive to the hyperparameter \( r \). However, the paper lacks a thorough investigation into how to optimally choose this parameter across different tasks and environments. More robust guidelines or heuristic methods for setting this parameter would strengthen the work.

- **Complexity of Algorithm Presentation:** While the algorithms RAGDP-VP and RAGDP-VE are explained, the presentation might be too complex for readers not deeply familiar with stochastic differential equations and diffusion models. Simplified explanations or additional illustrative examples could make the paper more approachable.

- **Elaboration on Vector Database Implementation:** The paper briefly mentions the use of FAISS for the vector database implementation but does not delve into the details of its configuration, indexing methods, and search efficiency. More information on these aspects would add clarity and allow for better reproducibility of the results.

**Questions:**

### Questions and Suggestions for the Authors

1. **Comparison with More Baseline Methods:**
   - **Question:** Have you considered comparing RAGDP with other state-of-the-art imitation learning methods that do not use diffusion models?
   - **Suggestion:** Including such comparisons would provide a clearer picture of RAGDP's relative advantages and limitations.

2. **Hyperparameter Sensitivity:**
   - **Question:** Can you provide more insights into how the hyperparameter \( r \) was chosen for various tasks? Is there a heuristic or automated method to set this parameter?
   - **Suggestion:** A detailed analysis or guideline on setting \( r \) would enhance the robustness and reproducibility of the method.

3. **Algorithm Explanation:**
   - **Question:** Could you simplify the presentation of the RAGDP-VP and RAGDP-VE algorithms, perhaps with illustrative examples or a more intuitive walkthrough?
   - **Suggestion:** Simplified explanations and additional illustrations would make the algorithms more accessible to a broader audience.

4. **Details on Vector Database Implementation:**
   - **Question:** Can you elaborate on the configuration and optimization of the FAISS vector database, including indexing methods and search efficiency?
   - **Suggestion:** Providing these details would help in understanding the practical aspects of implementing the vector database and ensuring reproducibility.

In summary, it appears that RAGDP tends more towards being an engineering optimization and might not have enough originality.

---

> ### Author Response · Authors · 2024-11-23
>
> Thank you very much for your comments. Here are the answers to the discussion questions.
>
> ### Q1:
>
> Have you considered comparing RAGDP with other state-of-the-art imitation learning methods that do not use diffusion models?
> Suggestion: Including such comparisons would provide a clearer picture of RAGDP's relative advantages and limitations.
>
> ### A1:
>
> We have benchmarked LSTM-based models with Transformer-based models (BeT [1]) and Behaviour Cloning datasets, with the Diffusion Model as the best performing model as a starting point [2]. Another extension of the Diffusion Models is the Consistency Models [3]. It is shown in the diagram in the Appendix C that the proposed method can also improve the performance of Consistency Models.
>
>
> ### Q2:
>
> Can you provide more insights into how the hyperparameter ( r ) was chosen for various tasks? Is there a heuristic or automated method to set this parameter?
> Suggestion: A detailed analysis or guideline on setting ( r ) would enhance the robustness and reproducibility of the method.
>
> ### A2:
>
> Determining the hyperparameters is a very important point. We have made measurements on this at two additional points - check Sec 5.5.
>
> ### Q3:
>
> Could you simplify the presentation of the RAGDP-VP and RAGDP-VE algorithms, perhaps with illustrative examples or a more intuitive walkthrough?
> Suggestion: Simplified explanations and additional illustrations would make the algorithms more accessible to a broader audience.
>
> ### A3:
>
> Diffusion Policy uses only the diffusion model of DDPM for its implementation. There are two main types of diffusion models: VP-SDE (Variance Preserving Stochastic Differential Equations) and VE-SDE (Variance Exploding Stochastic Differential Equations). The difference between the properties of each is the difference in Eq 3. The differences in their properties can be explained using the parameters $\alpha(\tau), \sigma(\tau)$ in Eq 3.
> In VP-SDE, the parameters are constrained by $\alpha(\tau)^2 + \sigma(\tau)^2 = 1$. This means that the magnitude of the noise and the action are determined by a ratio. Therefore, the action retrieved from the database is used to calculate the final output from the ratio of action and noise corresponding to the starting diffusion step $\tau_0$.
> In the case of VE-SDE, $\alpha(\tau) = 1$ is fixed. Therefore, there is no limit on the size of the action and noise. Therefore, the action taken from the database adds noise of a magnitude corresponding to the starting diffusion step $\tau_0$, and the output is obtained where this noise becomes smaller. Section 4.2 has been modified.
>
> ### Q4:
>
> Can you elaborate on the configuration and optimization of the FAISS vector database, including indexing methods and search efficiency?
> Suggestion: Providing these details would help in understanding the practical aspects of implementing the vector database and ensuring reproducibility.
>
> ### A4:
>
> Figure 3 has been updated to provide more detail. And code for a vector database using FAISS has been added to Appendix D. Information is stored in two files: a FAISS file for indexing and a Pickle file for storing action vectors.
>
> [1] Shafiullah, Nur Muhammad (Mahi) et al. “Behavior Transformers: Cloning k modes with one stone.” ArXiv abs/2206.11251 (2022): n. pag.
>
> [2] Chi, Cheng et al. “Diffusion Policy: Visuomotor Policy Learning via Action Diffusion.” ArXiv abs/2303.04137 (2023): n. pag.
>
> [3] Prasad, Aaditya et al. “Consistency Policy: Accelerated Visuomotor Policies via Consistency Distillation.” ArXiv abs/2405.07503 (2024): n. pag.

---

> > ### Author Response · Authors · 2024-12-03
> >
> > Dear reviewer,
> >
> > As the discussion period ends soon, we kindly ask for your engagement with our rebuttal.
> >
> > Thank you very much for the time and thoughtful comments so far.

---

### Official Review · Reviewer_FSfw · 2024-11-02

**Soundness:** 2
**Presentation:** 3
**Contribution:** 2
**Rating:** 5
**Confidence:** 4

**Summary:**

This paper proposes to improve the efficiency while preserve the effficacy of the diffusion-based policy generation, which is achieved by utilizing the retrieval.

**Strengths:**

The logic is clear and easy to follow.

**Weaknesses:**

There are several confusions that required to be answered:

1) Compared with other tasks, what are the special requriments for imitation learning on the utilization of retrieval-based diffusion?

2) Sec3.2 is confusing. Is mentioned prediction horizons T_p means steps within single diffusion? Or it means a sequence of actions in the imitation learning?

3) Given only the most similar are used for speeding up the diffusion process, what if the retrieval outputs a bad matching? Another thing is any possible that this retrieval operation degenerates the generalizability of the model?

4) It is not clear how the Encoder can be jointly trained with the policy generator, especially there is a matching algorithm involved. The details should be presented.

**Questions:**

Please refer to the Weaknesses.

---

> ### Author Response · Authors · 2024-11-23
>
> Thank you very much for your comments. Here are the answers to the discussion questions.
>
> ### Q1:
>
> Compared with other tasks, what are the special requirements for imitation learning on the utilization of retrieval-based diffusion?
>
> ### A1:
>
> The present method is proposed as a performance extension of the diffusion model. The key performance factors are accuracy, generation speed and robustness to the environment; Sec 5.2 and 5.3 show that speed can be increased without loss of accuracy. It has also been shown to improve accuracy when utilised with faster Consistency Models. Could you be more specific about the special requirements you indicate?
>
> ### Q2:
>
> Sec3.2 is confusing. Is mentioned prediction horizons $T_p$ means steps within single diffusion? Or it means a sequence of actions in the imitation learning?
>
> ### A2:
>
> This is unchanged from the Diffusion Policy. If the time step at time $t$ is $a_t$, then $A_t$ is a chunked action as $a_{t-T_o:t-T_o+T_p}$. Section 3.2 has been modified.
>
> ### Q3:
>
> Given only the most similar are used for speeding up the diffusion process, what if the retrieval outputs a bad matching? Another thing is any possible that this retrieval operation degenerates the generalizability of the model?
>
> ### A3:
>
> We believe that the key here is in the design of the database. The data for PH is less noisy for experts and more noisy for MH. In this experiment, the models trained on SQUARE-PH were allowed to search the SQUARE-MH database and the models trained on SQUARE-MH were allowed to search the SQUARE-PH database. As a result, models trained on SQUARE-PH tend to be less accurate when searched on SQUARE-MH data. However, models trained on SQUARE-MH tend to go down when searched on SQUARE-PH. Therefore, it is desirable that the training data for the model and the data in the vector database are similar. We also investigated how the results changed as the size of the database changed. - Appendix B.4 added.
>
>
> ### Q4:
>
> It is not clear how the Encoder can be jointly trained with the policy generator, especially there is a matching algorithm involved. The details should be presented.
>
>
> ### A4:
>
> The training of the encoders for the observed data $O_t$ remains unchanged from the Diffusion Policy. In the case of images, the encoders of the CNN are used to compute the vectors, while in the case of states, they are converted to 1D vectors for retrieval.
> First, when training, the diffusion model and encoder are trained together. Then, when the database is created, the encoders used in training are used to calculate the vectors in the case of images, while in the case of states, the vectors are converted to 1D vectors. Then, when inferencing, the vectors calculated once with the encoder or vectors with the dimension changed are used to search and retrieve the behavioral data. Figure 3 has been updated with more detail. Code for searching has been added in Appendix D.

---

> > ### Author Response · Authors · 2024-12-03
> >
> > Dear reviewer,
> >
> > As the discussion period ends soon, we kindly ask for your engagement with our rebuttal.
> >
> > Thank you very much for the time and thoughtful comments so far.

---

### Official Review · Reviewer_1zMY · 2024-11-03

**Soundness:** 3
**Presentation:** 3
**Contribution:** 2
**Rating:** 3
**Confidence:** 4

**Summary:**

This paper introduces RAGDP (Retrieve-Augmented Generative Diffusion Policy), which aims to improve the efficiency of diffusion policies without sacrificing accuracy. The main idea is to combine retrieval-based methods with diffusion models by storing expert observation-action pairs in a vector database. The paper presents two algorithms - RAGDP-VP and RAGDP-VE - which correspond to different types of diffusion models. The approach enables faster action generation while maintaining performance by leveraging retrieved expert demonstrations to inform the diffusion process.

**Strengths:**

1. RAGDP does not necessarily require additional training and can be integrated with existing approaches.

2. The paper is well-written and largely clear.

**Weaknesses:**

1. RAGDP seems sensitive to model selection and hyperparameters, but there's limited guidance on how this can be done without a reasonably exhaustive and task-specific search.

2. The effectiveness RAGDP depends heavily on the quality and coverage of the expert demonstration database, but this isn't thoroughly analyzed.

3. I don't find the vector database approach to be very scalable for very large datasets or high-dimensional action spaces.

**Questions:**

1. I'm not very convinced by the problem setting described by the authors. A number of works have already demonstrated diffusion policy works in real-world , real-time settings, with limited real-world data and limited latency problems [1][2][3]. Furthermore, the evaluation suite for RAGDP strictly relies on simulation tasks, whereas the authors describe their method in lines 76 and 77 as being "ideal for real-time robotic tasks." Finally, retrieval for control seems ill-advised, where small perturbations in the real world can deviate significantly from expert demonstrations, which often times are themselves suboptimal. Can the authors provide a more convincing portrayal for why one would want to use RAG in the diffusion policy context?

2. Related to the above question, but also more concretely:

i) How does the method perform with noisy or suboptimal expert demonstrations?

ii) What is the impact of the vector database size on performance?

iii) How sensitive is the method to the choice of embedding space for retrieval?

Without answering these questions, I remain unconvinced that RAG for DP is sensible.

3. The action dimensions for the tasks (as presented in Table 1) are all quite low (i.e. less than 10). How well does RAGDP do on tasks with much higher action dimensions (even just on simulation-based control tasks like those in Mujoco)?

---

> ### Author Response · Authors · 2024-11-23
>
> Thank you for your valuable points.  Here are the answers to the discussion questions.
>
> ### A1:
>
> About real-time settings:
>
> We appreciate this point as we want it to be clear that these models show potential in being used for real-world settings and that our method can further improve the performance. We modified the text to clarify that the method is more general and can help improve applications where the inference speed and accuracy are important.
>
>
> ### A2:
>
> 1. About noisy or suboptimal expert demonstrations (Robustness for generating quality):
>
> The experiments conducted in Appendix B.3 might be helpful to answer your question. Specifically, we evaluated the SQUARE-PH and SQUARE-MH datasets, where SQUARE-PH consists of proficient demonstrations, and SQUARE-MH includes a mix of proficient and suboptimal demonstrations. We perform two experiments: i) Training the base model on high quality demonstrations provided by experts and using a knowledge base on demonstrations of mixed quality and ii) Training the base model on a mixed quality of demonstrations with a knowledge base on high quality demonstrations. These experiments show both the effect of a distribution shift (as the model trained on mixed data and trained on expert data will behave differently) of the demonstrations as well as suboptimal demonstrations.
>
> Table 5 (Base model trained on expert data and knowledge base trained on mixed data) shows that there is a decrease in performance with suboptimal demonstrations albeit still relatively high considering the harsh scenario of mismatched and suboptimal demonstrations. This can further be mitigated by setting a conservative $r$.
>
> Table 6 (Base model trained on mixed data and knowledge base on expert data) shows a much smaller decrease in accuracy even with the difference in demonstrations.
> We hope that this experiment shows that the trade-off is acceptable even in sub-optimal conditions and that it can be proactively mitigated by picking the $r$ hyperparameter.
>
>
> 2. About vector database size:
>
> We investigated the relationship between accuracy and retrieval distance as the size of the database varied. The results, shown in Figure 12, indicate that retrieval distance decreases as the database size increases. However, this change in retrieval distance does not significantly affect accuracy. This suggests that the accuracy of the method is robust to retrieval distance. Instead, the quality of the data stored in the database appears to play a more critical role than its quantity. Supporting this observation, prior studies [3] have demonstrated that retrieving high-quality data during training can improve accuracy, even when the dataset size is relatively small.
>
> 3. About embedding spaces:
>
> The use of VAEs to embed observations and search in the embedding space, as in study [3], was also initially considered by us. However, the accuracy results did not change from the current method. We have also adopted the current retrieval method because the proposed method is easier to implement without the need to train additional models.
>
> ### A3:
> Additional experiments were conducted with Transport, which has twice the action dimension in the behavior cloning dataset. Similar results were obtained here - check Table 4, Figure 9.
>
>
> [1] Prasad, Aaditya et al. “Consistency Policy: Accelerated Visuomotor Policies via Consistency Distillation.” ArXiv abs/2405.07503 (2024): n. pag.
>
> [2] Tangkaratt, Voot et al. “VILD: Variational Imitation Learning with Diverse-quality Demonstrations.” International Conference on Machine Learning (2019).
>
> [3] Du, Maximilian et al. “Behavior Retrieval: Few-Shot Imitation Learning by Querying Unlabeled Datasets.” ArXiv abs/2304.08742 (2023): n. pag.

---

> > ### Comment · Reviewer_1zMY · 2024-12-02
> >
> > I thank the authors for their response. In summary, I had two main points of contention with the work:
> >
> > 1. Diffusion policy (DP) has been documented in prior art to work out of the box in real-world, real-time settings, with limited latency problems, even given limited real-world data [1][2][3] (references were missing in my original comment so I'll provide them here again). This makes the goal of using RAG to improve the efficiency of DP to be a misguided one from my perspective.
> >
> > 2. The scalability, generalizability and robustness of the method were all unclear to me.
> >
> > In general, I do not feel like these two questions were adequately addressed in the response. In particular, the authors answer in A1 does not convince me that adding RAG to DP solves any fundamental issues of efficiency that DP only suffers from. I can maybe be convinced that under highly dynamic settings with limited feedback, such efficiency problems might emerge, but from the experimental suite the authors provide in the current draft (especially in a simulation-only setting), this narrative is not supported.
> >
> > The authors' responses in A2/A3 are also unconvincing. Specifically, the error bars in Figure 12 are almost completely overlapping each other, so I'm not sure there's enough experimental signal here to conclude that "accuracy is not considered to be directly related to retrieve distance." Moreover, the authors claim to have run experiments with VAE embeddings. Firstly, it would be nice to show these results, even if they are not better, or outright worse. Secondly, my question was not so much about the particular model or loss function used to learn embeddings, but rather a more general question about the sensitivity of retrieval to embedding spaces, and how such spaces could be vastly different for different tasks. It seems investigation along this axis was not performed in the current draft. I recommend the authors spend more time reorganizing the experimental exposition to better answer all the questions under 2.
> >
> > In summary, my score remains in favor of rejection.
> >
> > [1] "Scaling Up and Distilling Down: Language-Guided Robot Skill Acquisition." Ha et al. CoRL 2023.
> >
> > [2] "Octo: An Open-Source Generalist Robot Policy." Octo Model Team et al. arXiv 2024.
> >
> > [3] "Learning Visuotactile Skills with Two Multifingered Hands." Lin et al. arXiv 2024.

---

### Official Review · Reviewer_kPzD · 2024-11-04

**Soundness:** 3
**Presentation:** 3
**Contribution:** 3
**Rating:** 6
**Confidence:** 2

**Summary:**

RAGDP addresses the slow performance of sampling from diffusion policy model by incorporating a vector database containing pairs of expert observations and actions. This database is used to retrieve relevant actions during the generation process, which helps maintain high accuracy while speeding up policy inference.

**Strengths:**

The paper introduces an innovative method, RAGDP (Retrieve-Augmented Generative Diffusion Policy), which integrates retrieval-augmented generation into the realm of Diffusion Policies for imitation learning. This approach is original in that it applies the concept of retrieval augmentation, typically used in large language models, to enhance the efficiency of Diffusion Policies. By leveraging a vector database containing expert observation-action pairs, the authors create a system that retrieves high-similarity actions during inference to accelerate the generation process without sacrificing accuracy

The paper is generally clear and well-structured, providing readers with sufficient background on Diffusion Policies, stochastic differential equations, and retrieval-augmented methods

The paper evaluates this approach on four common and widely used samplers and show performance gains across all of them.

**Weaknesses:**

* Adding RAG pipelines to existing frameworks is comparatively engineering heavy compared to simpler methods, but the results are strong enough that they might warrant the extra effort.
* The method is sensitive to hyper parameters as the authors admit in the conclusion.

**Questions:**

* A lot of white space is wasted in Figure 3

---

> ### Author Response · Authors · 2024-11-23
>
> We thank the reviewer for the valuable feedback which gives us the opportunity to improve the paper. We have made changes to the paper accordingly by clarifying the implementation as well as providing additional experiments.
>
> 1. Difficulty of adding RAG pipeline into existing method:
>
> We appreciate the reviewer's concern about the engineering effort. To clarify, integrating RAGDP is relatively straightforward as it leverages existing diffusion models without requiring changes to their core architectures. Basically, to summarize the engineering process:
>
> Stage 1: The training data is normalized and stored as a vector database using FAISS.
>
> Stage 2: At inference, retrieved actions are incorporated into the diffusion denoising process using a few modifications to the sampling algorithm.
>
> To make this clearer, we have updated Figure 3 to include more implementation details, and we added an explanation in Section 4.1 highlighting the modularity of the retrieval component. This modularity ensures that RAGDP can be easily adopted in conjunction with other diffusion-based approaches.
>
>
>
> 2. Choosing the optimal hyperparameter:
>
> We agree with the reviewer that selecting appropriate hyperparameters is important. To give a more clear picture of the robustness of the hyperparameter choice, we have conducted additional experiments varying the hyperparameter r. The results, provided in Section 5.5, show that the method maintains robustness across a reasonable range of hyperparameter values, not showing a noticeable decrease until $r > 0.75$.

---

> > ### Author Response · Authors · 2024-12-03
> >
> > Dear reviewer,
> >
> > As the discussion period ends soon, we kindly ask for your engagement with our rebuttal.
> >
> > Thank you very much for the time and thoughtful comments so far.

---

### Meta-Review · Area_Chair_CbbH · 2024-12-18

**Metareview:**

The paper presents a new approach called Retrieve-Augmented Generative Diffusion Policy (RAGDP), which combines retrieval-augmented generation with diffusion policies in the context of imitation learning. The authors claim this integration can speed up the learning process without compromising on accuracy.

The strengths of the paper lie in its innovative approach to enhancing the efficiency of diffusion policies, which is a notable contribution to the field. The paper is also commended for its clarity and thorough explanation of the methods involved.

However, there are several weaknesses that need to be addressed. The novelty of the approach is limited, as retrieval-augmented techniques have been explored in other domains. Reviewers also raised concerns about the paper's sensitivity to hyperparameters and its scalability, especially with larger datasets or high-dimensional action spaces. There's also an apparent lack of analysis on how the quality of the expert demonstration database affects the results.

The decision to reject is primarily due to these weaknesses, particularly the limited novelty, the issues with hyperparameters and scalability, and the insufficient analysis on the influence of expert demonstration quality. These concerns were highlighted by multiple reviewers and were not satisfactorily resolved during the rebuttal phase.

**Additional Comments On Reviewer Discussion:**

During the discussion, Reviewer 1zMY expressed concerns regarding the sensitivity of the system to hyperparameters, and Reviewer FSfw pointed out potential scalability issues. The authors responded with additional details on implementation and extra experiments to showcase robustness across different hyperparameter values. However, these responses did not fully address the concerns, particularly those of Reviewer 1zMY related to hyperparameter sensitivity and FSfw's scalability issues.

In weighing these concerns against the authors' responses, it became clear that the issues raised during the review process remained significant and were not adequately resolved in the rebuttal. While the authors made an effort to address the reviewers' feedback, their responses fell short of fully convincing the reviewers and myself of the robustness and scalability of the proposed approach.

In conclusion, while the paper does have some strong points, the significant concerns regarding novelty, scalability, hyperparameter sensitivity, and a lack of thorough analysis on the influence of expert demonstration database quality have led to the recommendation to reject the submission.

---

### Decision · Program_Chairs · 2025-01-22

Reject